

# Emerging global freshwater challenges unveiled through observation-constrained projections

Fei Huo[1], Yanping Li[1], Zhenhua Li[1]

[1]Department of Physics and Astronomy, Western University, London, N6G 2V4, Canada.

*Correspondence to*: Yanping Li (yli4972@uwo.ca)

**Abstract.** Future hydrological projections exhibit significant discrepancies among models, undermining confidence in the predicted magnitude and timing of hydrological extremes. Here we show that observation-constrained changes in global mean terrestrial water storage (TWS), excluding Greenland and Antarctica, could be approximately 83 mm lower than raw projections from the Inter-Sectoral Impact Model Intercomparison Project phase 3b by the end of this century under both
low- and high-end future forcing scenarios. Notably, the 95th percentile upper bounds are substantially reduced from 2 mm to -96 mm under the low-emissions scenario and from 8 mm to -105 mm under the high-emissions scenario, revealing a notable overestimation of global freshwater availability in the raw model projections. Global models are intricate process representations, making it challenging to isolate causes of their differences with observations. However, by leveraging the emergent constraint (EC) methodology and inter-model spread to empirically adjust biases against observations, we produce
more robust projections of future TWS changes than conventional, unconstrained approaches. EC-corrected projections indicate a significant decrease in freshwater availability, further exacerbating existing water stress worldwide if global water demand remains stable or continues to rise. Our findings pinpoint the urgent need to reduce model uncertainties and enhance the reliability of future hydrological projections to better inform water resource management and climate adaptation strategies.

**1 Introduction**

Terrestrial water storage (TWS) encompasses all water stored on and beneath the land surface, representing the net balance of precipitation, evapotranspiration, and runoff (Getirana et al., 2017; Rodell & Famiglietti, 2001). As a critical element of the hydrological cycle, TWS plays a key role in regulating freshwater availability (Rodell & Famiglietti, 2001) and Earth's energy budget (Getirana et al., 2017), supporting freshwater ecosystems (Tapley et al., 2019; Wu et al., 2024), influencing
biogeochemical cycles (Rodell et al., 2018), driving socioeconomic development (Scanlon et al., 2023; Vörösmarty et al., 2000), and mitigating sea level rise by enhancing continental water storage (Tapley et al., 2019).

A warming climate impacts TWS by accelerating the hydrological cycle through enhanced evapotranspiration and by modifying global precipitation patterns (A. Dai et al., 2018). These changes exacerbate freshwater scarcity under climate





change, highlighting the crucial need for accurate future projections. However, the discrepancies among models in simulating historical hydrological budgets lead to substantial differences in their projections of extreme hydrological events (Herrera-Estrada et al., 2017; Trenberth et al., 2003; Vogel et al., 2018), which in turn reduce confidence in the predicted magnitude and timing of these extremes. These discrepancies can be attributed to various factors, including uncertainties in climate forcing (Scanlon et al., 2018), the absence of key components such as surface water storage, groundwater storage,

and human interventions in most land surface models (LSMs), as well as limited storage capacities within both LSMs and global hydrological models (GHMs).

The uncertainty of future projections can be constrained using the emergent constraint (EC) approach (Brient, 2020; Hall et al., 2019), which identifies potential physical relationships between the observable historical climate and future change

across global models. While a large number of ECs have been proposed for global mean temperature changes and other hydro-climatic variables (Bowman et al., 2018; Brient, 2020; Hall et al., 2019; Petrova et al., 2024; Shiogama et al., 2022), the potential constraints on global mean changes in terrestrial water storage have yet to be thoroughly explored. To fill this gap, here we examine EC relationships using hydrological simulations from multiple LSMs and GHMs to identify the physical mechanisms underlying changes in TWS. By combining the proposed EC with historical observations from Gravity

Recovery and Climate Experiment (GRACE) satellites (Tapley et al., 2019; Velicogna et al., 2020), we successfully constrain future TWS changes and obtain EC-corrected late century projections.

## 2 Data and methods

### 2.1 Observations

Global observations (excluding Greenland and Antarctica) for the period 2004–2023 were derived from GRACE satellites.

Because research has demonstrated that GRACE data processing in terms of mass concentration (mascon) solutions results in higher correlations with in situ data compared to spherical harmonic solutions (Watkins et al., 2015), we utilized all three available GRACE mascon solution datasets (i.e., JPL RL06.3M v04, CSR RL0603M, and GSFC mascon RL06 v1.0), produced by the Jet Propulsion Laboratory (Watkins et al., 2015), the Center for Space Research (Save et al., 2016), and the Goddard Space Flight Center (Luthcke et al., 2013), respectively, to estimate uncertainties. To ensure our results are robust

across processing approaches, we also incorporate four spherical-harmonic products (i.e., JPL RL06, CSR RL06, GFZ RL06, and COST-G RL01). To address missing months in the GRACE datasets, we employed linear interpolation to estimate and fill these gaps.

### 2.2 Global models and climate forcing

Five LSMs and GHMs (see details in Table S1) from the Inter-Sectoral Impact Model Intercomparison Project phase 3b

(ISIMIP3b; https://protocol.isimip.org/#/ISIMIP3a), were employed to assess historical and future relationships in TWSAs





(see details below). These projections were based on climate forcing from the Coupled Model Intercomparison Project Phase 6 (CMIP6) (Eyring et al., 2016). Climate forcing data were sourced from five general circulation models (GCMs) under three scenarios: historical climate (HIST, 1850–2014), a low greenhouse gas (GHG) emissions scenario (SSP1-2.6), and a high GHG emissions scenario (SSP3-7.0). These scenarios were chosen to maximize the inclusion of available models. The

historical climatology (2004–2023) was constructed by combining the end of the HIST run with the beginning of the SSP1-2.6 simulation, following the approach adopted in previous TWS studies using the ISIMIP datasets (for example, ref. (Pokhrel et al., 2021)). SSP2-4.5, a "middle of the road" scenario, would be the most appropriate for extending the historical period due to its alignment with historical socioeconomic trajectories (Fricko et al., 2017). Nevertheless, only a limited number of models provide the SSP2 forcing in the ISIMIP datasets. To ensure a robust analysis, the proposed ECs were

validated using an ensemble of simulations from eight ISIMIP2b models (Table S2). These simulations were based on climate forcing from four CMIP5 GCMs under three scenarios: historical climate (HIST, 1861–2005), the RCP2.6 scenario, and the RCP6.0 scenario. All outputs were provided at a monthly temporal resolution and on a $0.5° × 0.5°$ global grid. Monthly data were then regridded to a common $1° × 1°$ global grid. The ensemble members (i.e., outputs from each GHM or LSM driven by different climate forcings) were compared with GRACE data. To ensure consistency, we computed TWSAs

at each grid point relative to GRACE's baseline period of 2004–2009. This aligns all datasets to a common reference, conducting direct comparison of observed and simulated TWSAs as deviations from the same climatological mean.

## 2.3 Emergent constraint approach and calibration

To implement the EC framework, we begin by identifying statistically significant relationships between annual global mean changes in TWS, **y**, and historical annual global mean TWSA climatologies, **x**, across a variety of global models. To ensure

the robustness of our EC results, we also used linear trends of historical TWSAs as alternative predictors. Specifically, the long-term trend in historical annual TWSAs at each grid point was estimated using ordinary least squares regression. A linear regression model was employed to depict the EC relationship between **x** and **y**. The variable **x** was replaced with the mean of historical observations, and the mean of EC-corrected changes was derived using the regression model. We also calibrated the projected future changes by applying the EC relationship to each grid cell, resulting in the spatial distribution

of EC-corrected projections along with the corresponding biases.

To estimate the uncertainty of the calibrated future changes, a stationary bootstrap method (Brient, 2020; Politis & White, 2004) was applied. Bootstrapping was performed 500 times, generating 500 regression line samples. This approach robustly quantifies the sampling uncertainty without requiring assumptions about the underlying probability distributions. Following

the method of Brient and Schneider (Brient & Schneider, 2016), confidence intervals for the calibrated future changes were estimated by projecting the observed value $x_o$ from each GRACE dataset (using all three mascon solutions) onto the generated 500 regression line samples.



Spearman's rank correlation coefficients were calculated for TWS-related variables to investigate the physical mechanisms

underlying the EC relationship. These variables were derived from simulations under the historical and SSP3-7.0 scenarios. To understand the spatial distribution of differences in future changes, we categorized the six driest and six wettest ensemble members based on their historical climatology of annual global mean TWSAs (Table S1). For simplicity, we refer to models with historically low and high simulated TWSAs as "dry" and "wet" models, respectively. The observed discrepancies are likely due to differences in how models represent storage compartments and account for human influences. The driest

models produced resent-day global mean TWSAs that closely align with values derived from mascon solutions, while the wettest models appeared to overestimate the historical climatology (Fig. 1).

## 3 Results

### 3.1 Proposed emergent constraint derived from historical climatology

Our approach follows established EC protocols by relating an observable historical metric (that is, area-weighted global

mean of annual terrestrial water storage anomaly (TWSA) climatology) to a targeted future change (global mean of end-of-century annual TWS change) across an ensemble of models. This strategy is consistent with prevailing EC studies (for example, (Cai et al., 2025; P. Dai et al., 2024; Kim et al., 2023; Petrova et al., 2024)), which often relate present-day climatological states (or trends when they offer stronger physical connections) to future changes rather than link historical and future trends directly. To align GRACE observations with model outputs, we first compute TWSAs at each grid point

relative to GRACE's default baseline period of 2004–2009. Anchoring all datasets to this common reference ensures that both historical and future TWSAs (whether from observations or models) are directly comparable as deviations from the same climatological mean. Consequently, because both historical and future TWSAs share this baseline, their difference naturally represents changes in TWS (hereafter referred to as "TWS change") rather than changes in anomalies. In fact, our approach to calculating TWS change has shown results comparable to the conventional method (cf. Fig. 2c,d in this article

with Fig. 1b,d in (Pokhrel et al., 2021)). Finally, we derive a TWSA climatology by averaging monthly anomalies for each calendar year and then computing the multi-year mean over the period of interest.

We analyze historical and future relationships in TWSAs using multi-model hydrological simulations of 25 ensemble members from the Inter-Sectoral Impact Model Intercomparison Project (Warszawski et al., 2014) phase 3b (ISIMIP3b), and

31 ensemble members from the ISIMIP2b (Frieler et al., 2016) (Methods and Supplementary Tables 1 and 2). Future projections are based on climate forcing from the Coupled Model Intercomparison Project phase 6 (CMIP6) (Eyring et al., 2016) and CMIP5 (Taylor et al., 2012), respectively. To maximize the inclusion of available models and compare with previous studies (for example, (Pokhrel et al., 2021)), we select a low-emissions scenario (Shared Socio-economic Pathway



(SSP)1-2.6/Representative Concentration Pathway (RCP)2.6) and a high-emissions scenario (SSP3-7.0/RCP6.0). Significant
positive correlations ($R > 0.99$ for both ISIMIP2b and ISIMIP3b models) are found between historical and late century annual area-weighted global mean TWSAs, irrespective of the emissions scenario (Fig. S1). These findings indicate that models simulating higher TWSAs in the historical climate tend to predict higher TWSAs in the future, in agreement with the "wet gets wetter" atmospheric response to warming identified globally (Held & Soden, 2006) and over land (Petrova et al., 2024) in the CMIP models. Various hydrological datasets also corroborate this "wet gets wetter" signal over water-sufficient
lands (Greve et al., 2014; Kumar et al., 2015; Markonis et al., 2019). GHMs and LSMs are developed based on distinct numerical frameworks and parameterizations, leading to structural diversity in how they represent key storage components. This structural diversity such as lack of surface water and groundwater storage compartments and human intervention in most LSMs; see Supplementary Tables 1 and 2) is responsible for inter-model differences (Scanlon et al., 2018), which may lead to the "wet gets wetter" pattern. For instance, under the SSP1-2.6 scenario, MIROC-INTEG-LAND simulates relatively
low TWSA climatologies in both historical and future periods, whereas JULES-W2 produces much higher values under the same conditions (Fig. S1). Our methodology does not assume uniform storage physics; instead, it leverages the multi-model spread to empirically constrain biases against GRACE observations.

As shown in Fig. 1, the magnitude of future TWS increases is positively correlated with the historical TWSA climatology.
The Spearman's rank correlation coefficients between the historical TWSA climatology and mid-century TWS changes are 0.79 and 0.70 ($p < 0.01$) for the ISIMIP2b and ISIMIP3b ensembles, respectively, under the low-emissions scenario. Under the high-emissions scenario, statistically significant correlations of 0.61 and 0.71 ($p < 0.01$) are observed for the ISIMIP2b and ISIMIP3b ensembles, respectively. For late century TWS changes, significant correlations with historical TWSA climatology persist across both ISIMIP2b and ISIMIP3b ensembles. This robust relationship provides a basis for
constraining future TWS changes (Methods). It is worth noting that several modeling factors including initial conditions, structural differences (especially the diversity of storage compartments), human intervention, and calibration can contribute to the large inter-model spread and thus the resulting discrepancies with GRACE observations.



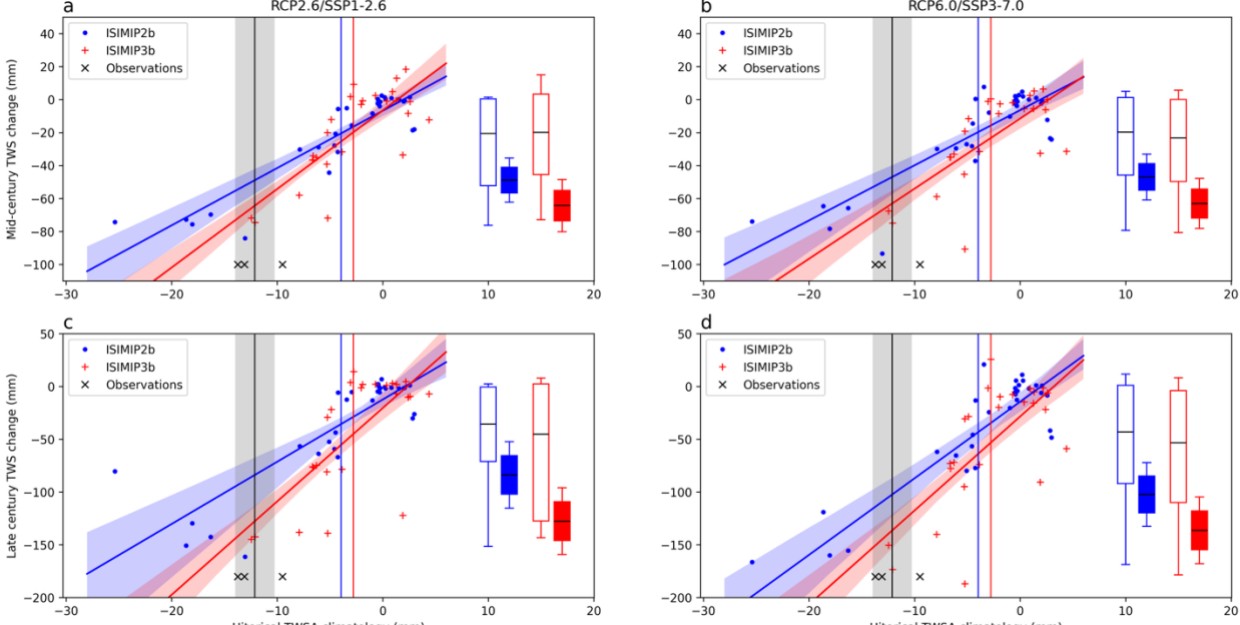

**Figure 1: Inter-model relationships between historical (2004–2023) climatologies and future changes at mid-century (2040–2059; upper panels) and late century (2080–2099; lower panels) from ISIMIP2b (blue) and ISIMIP3b (red) models under the RCP2.6/SSP1-2.6 (a,c) and RCP6.0/SSP3-7.0 (b,d) scenarios. Dots and crosses represent global (excluding Greenland and Antarctica) averages of TWSAs from ensemble members. Blue and red lines represent linear regression fits, with 90% confidence intervals estimated through bootstrapping. Blue and red vertical lines mark the ensemble mean. Black vertical lines indicate the**
**average of GRACE observations (black cross), and grey shading represents the standard deviation. Box plots indicate the mean (black line), 66% (box), and 90% (whisker) confidence intervals of future TWS changes before (empty box) and after (filled box) applying observational constraints.**

After applying the EC calibration (Methods), mid-century global mean TWS changes are reduced by 44 mm and 40 mm

compared to the raw projections from the ISIMIP3b ensembles under the low- and high-end forcing scenarios, respectively (Fig. 1, upper panels). For late century projections, EC-corrected changes could be ~83 mm lower than the raw estimates from the ISIMIP3b ensembles irrespective of the forcing scenario (Fig. 1, lower panels), highlighting potentially lower global freshwater availability than initially indicated by the ISIMIP3b models. Furthermore, the EC correction constrains the discrepancies of late century TWS changes by 63% for the SSP1-2.6 scenario and 69% for the SSP3-7.0 scenario.

Specifically, the upper bound (95th percentile) is reduced from 2 mm to −96 mm under the low-end forcing scenario and from 8 mm to −105 mm under the high-end forcing scenario, indicating an initial overestimation of global freshwater availability in the raw ISIMIP3b ensemble projections. Global models are sophisticated process representations, making it challenging to isolate causes of their differences with GRACE (Haddeland et al., 2011). However, by leveraging the EC methodology and inter-model spread in water storage partitioning against GRACE data, we produce more robust projections

of future TWS changes than conventional, unconstrained approaches. Note that the reduction in ensemble uncertainty shown



in our study is unexpectedly greater than that reported in previous EC-related research (e.g., refs. (Brient, 2020; Brient & Schneider, 2016; Petrova et al., 2024)). This discrepancy arises because GRACE products used for EC correction are all derived from a single source—the GRACE satellites—resulting in an underestimated uncertainty for global mean TWSA observations compared to other multi-source observational variables, such as air temperature and precipitation. EC-corrected changes based on the ISIMIP2b models show similar results to those from the ISIMIP3b models, albeit with higher ensemble averages, which can be attributed to the shallower slopes observed in their linear regression relationships. Although here we illustrate results using three mascon-based GRACE datasets, extending the analysis to include four spherical-harmonic solutions yields EC-corrected projections that remain robust but produces slightly higher mean TWS changes at the end of this century (Supplementary Fig. 7). We also analyze the past-future EC relationship using linear trends of historical TWSAs as potential predictors (Fig. S2, lower panels). The corresponding results closely align with those obtained using historical TWSA climatology as predictors, showing identical ensemble averages and consistent 5–95% ranges. Subsequent analyses are based on the past-future EC relationship with historical TWSA climatology as predictors, due to the current models' inability to accurately simulate the large regional TWSA trends observed by GRACE (Scanlon et al., 2018). This limitation may reduce the reliability of the spatial patterns in future TWSA change projections presented below.

## 3.2 Spatial patterns in future TWSA change projections

By the end of the 21st century, TWSAs are projected to decrease considerably across several regions under the SSP1-2.6 scenario, including the southern United States, Mexico, northwestern South America, both coasts of the strait of Gibraltar, the majority of Central, West, and South Asia, as well as North China (Fig. 2a). Under the SSP3-7.0 scenario, although the spatial pattern of TWS changes is similar, the signal becomes much stronger, especially in the low latitudes (Fig. 2b). The future changes projected by the ISIMIP2b ensembles are consistent with those from the ISIMIP3b ensembles across most land areas, with pattern correlations of 0.44 and 0.55 under the low- and high-end forcing scenarios, respectively. However, the smaller magnitude of the latitudinal mean in the ISIMIP2b projections indicates a weaker signal of TWSA change compared to the ISIMIP3b projections. Although this study uses an unweighted ensemble mean and a slightly different period to represent the late century, the spatial distributions closely align with those reported by ref. (Pokhrel et al., 2021) The primary distinction lies in the latitudinal mean, which reveals a much steeper decline in TWSAs over the northern midlatitudes irrespective of the emissions scenario in our analysis (Fig. 2c,d).



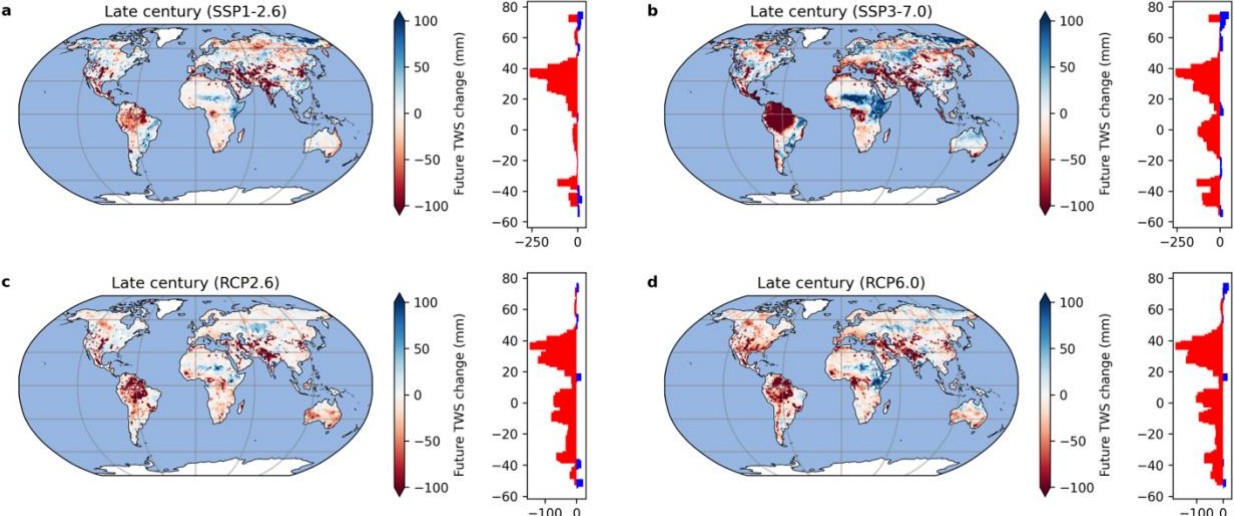

**Figure 2: Late century (2080–2099) multi-model ensemble averages of TWS changes under the SSP1-2.6/RCP2.6 (a,c) and SSP3-7.0/RCP6.0 (b,d) scenarios, shown relative to historical (2004–2023) climatologies. The accompanying histograms on the right indicate zonally averaged TWSAs for each scenario.**

To provide accurate regional-scale projections of future TWS changes, we obtained EC-corrected maps (Fig. 3a,c) by calibrating the ISIMIP3b outputs against observed TWSA climatologies at each grid cell (Methods). The proposed EC correlations are robust ($R > 0$ and $p < 0.05$) across ~26% of global land areas (excluding Greenland and Antarctica) under both scenarios, based on the ISIMIP3b ensembles. In contrast to the raw ISIMIP3b outputs (Fig. 2a,b), the EC-corrected maps show similar patterns, such as pronounced TWSA declines in the northern midlatitudes, though with greater magnitudes under both the low- and high-emissions scenarios, as highlighted in the histograms of the latitudinal mean (Fig. 3a,c). The similarity in spatial patterns between the raw model outputs and the EC-corrected projections, along with the bias maps (Fig. 3b,d), corroborates our global analysis: observation-constrained TWS changes (ensemble mean) shift away from zero after the EC correction (Fig. 1). Under both emissions scenarios, the ISIMIP3b outputs show a notable overestimation of TWSAs across mid- and high latitudes in the Northern Hemisphere, including regions such as the Northwest Territories in Canada, the Southwestern United States, much of the Middle East, the Danube River Basin, Siberia, and North China. This overestimation is consistent with previously reported underestimations of decreasing TWSA trends in models over Northern Hemisphere river basins compared to satellite observations (Scanlon et al., 2018), which may consequently lead to overestimated projections of future freshwater availability in these regions. Conversely, considerable underestimations of TWSAs occur in the Amazon and the Murray Basin in southeastern Australia, aligning with models' tendency to underestimate increasing TWSA trends in humid regions or nonirrigated basins (Scanlon et al., 2018).





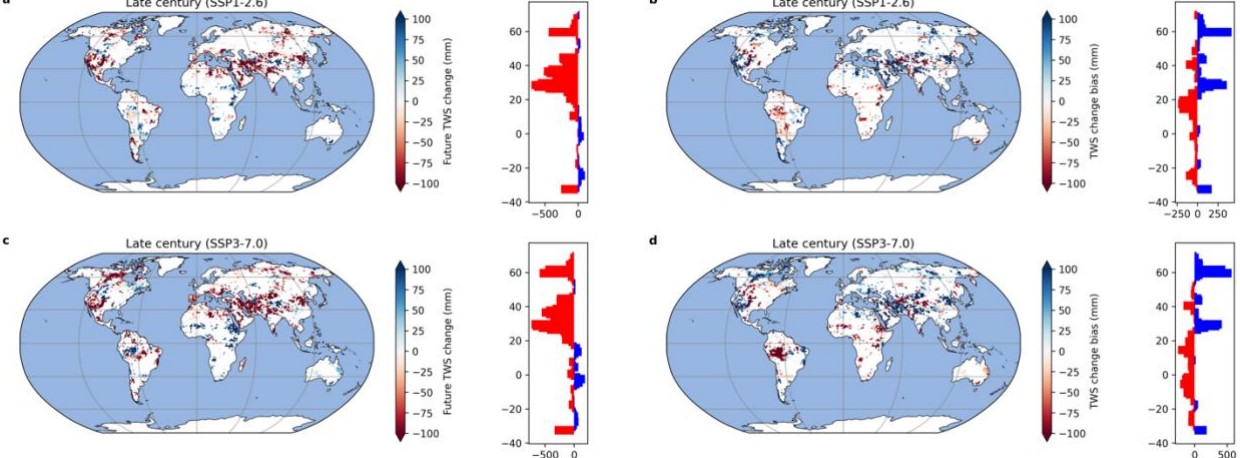

**Figuire 3: a,c,** Late century (2080–2099) EC-corrected TWS changes under the SSP1-2.6 and SSP3-7.0 scenarios, shown relative to historical (2004–2023) climatologies. **b,d,** Biases in projected late century TWS changes (raw model outputs minus EC-corrected values). Only regions with statistically significant positive EC correlations ($R > 0$ and $p < 0.05$) are shown. The histograms on the right represent zonally averaged values, with data shown only for latitudes between 70°N and 35°S due to sparse coverage outside this range.

### 3.3 Underlying physical processes of the emergent relationship

It is crucial to elucidate the physical mechanisms linking historical and future variability in the EC relationship (Caldwell et al., 2014; Hall et al., 2019; Schlund et al., 2020). Under the SSP3-7.0 scenario, significant positive inter-model correlations ($p < 0.05$) are found between local precipitation and TWS changes over most regions globally (Fig. 4a). Similarly, positive correlations are evident between local precipitation and other TWS-related variables, such as evapotranspiration and total runoff (Fig. 4b,c). These results are consistent with established physical understandings, affirming that local precipitation changes strongly correlate with TWS changes. More importantly, they suggest that models projecting higher precipitation changes tend to predict larger TWSA increases. This finding highlights the transfer of the "wet gets wetter" atmospheric response to warming into future hydrological projections through precipitation forcing. Actually, climate models predicting higher warming trends often anticipate greater precipitation increases as a result of thermodynamics (Emori & Brown, 2005; Shiogama et al., 2022). At local scales, atmospheric warming-induced increases in water vapor enhance precipitation in wet regions and reduce it in dry regions (Chou et al., 2013; Held & Soden, 2006).



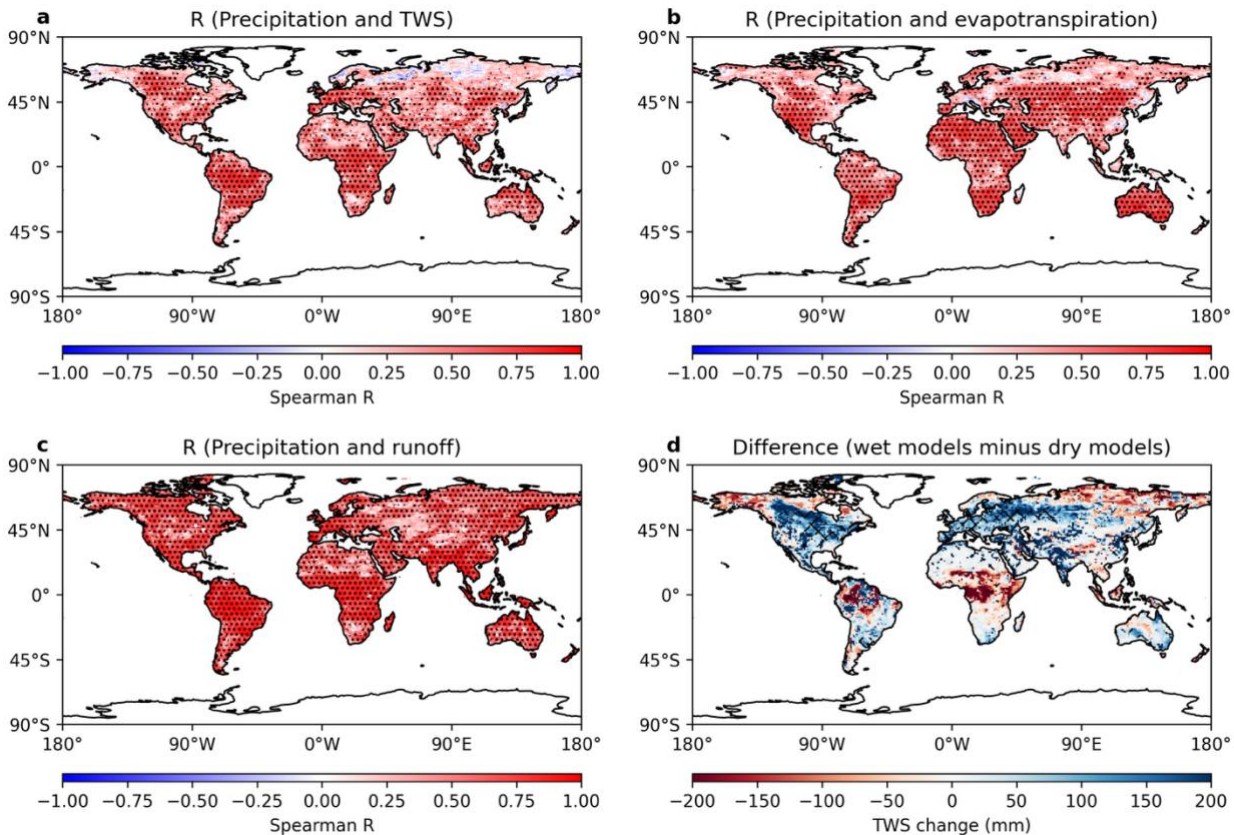

**Figure 4: Inter-model Spearman's rank correlations between late century (2080–2099) precipitation changes and changes in TWS (a), evapotranspiration (b), and total runoff (c) under the SSP3-7.0 scenario. Black stippling marks regions of statistical significance ($p < 0.05$). d, Differences in late century changes for TWSAs (the wettest minus the driest models). Black hatches indicate statistically significant differences at the 5% level, as determined by Welch's $t$-test. A permutation test with 100 random permutations was conducted to estimate the $p$-values.**

To further explore inter-model variations, we examine differences in TWS-related variables between the wettest and driest ensemble members based on their historical TWSA climatologies (Fig. 1, horizontal axis values; Methods). Significant differences in future TWS changes emerge, particularly in northern midlatitudes such as North America and Europe, where the driest ensemble members predict substantially more severe TWSA reductions compared to the wettest ones (Fig. 4d). Similar patterns of inter-model correlations and subdued precipitation changes are also found under the SSP1-2.6 scenario (Fig. S5).

**4 Discussion**

The EC relationship between historical TWSA estimates and future changes is statistically significant on both global and regional scales, especially across approximately 26% of global land areas. However, the reliability of proposed ECs could be





compromised due to the lack of independence among climate models (Brient, 2020; Caldwell et al., 2014). Climate models are frequently derived from one another (Knutti et al., 2013). This challenge becomes more pronounced in future hydrological projections. LSMs and GHMs often share structural similarities in simulating water storage compartments and human water use sectors (Telteu et al., 2021). Moreover, their climate forcings could also derive from climate models with

insufficient diversity. This is reflected in the skewed distribution of global averages of historical TWSA climatologies in the ISIMIP models before the EC correction (Fig. 1, empty box plots). These findings underscore the critical need to ensure diversity in global models as well as in their climate forcings, particularly in water-resources-focused projects like the ISIMIP.

In conclusion, our observation-constrained results highlight that warming-induced reductions in TWS translate into diminished freshwater availability on both global and regional scales. This is especially evident in mid- and high-latitude regions of the Northern Hemisphere, where low historical TWSA climatologies are prevalent. Compared to the raw ISIMIP3b projections, our constrained estimates indicate an average TWS decrease of roughly 83 mm, revealing a significant overestimation of future water availability in both GHMs and LSMs. This overestimation elevates the risk of

basins being underprepared for actual supply conditions, as uneven water gaps are projected to widen under warming scenarios (Rosa & Sangiorgio, 2025). Accordingly, there is an urgent need to reduce model uncertainties through robust observational constraints and enhance diversity among GHMs and LSMs, thereby improving the reliability of future hydrological projections for informed water resource management and climate adaptation.

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

**Acknowledgments**

Y.L. acknowledges support from a Natural Sciences and Engineering Research Council of Canada (NSERC) Discovery Grant.

**Data availability**

All ISIMIP output and climate forcing data are available at https://data.isimip.org. GRACE products can be obtained from https://grace.jpl.nasa.gov/data/get-data/jpl_global_mascons/, https://www2.csr.utexas.edu/grace/RL05_mascons.html, https://earth.gsfc.nasa.gov/geo/data/grace-mascons, https://podaac.jpl.nasa.gov/dataset/GRACEFO_L2_JPL_MONTHLY_0063, https://podaac.jpl.nasa.gov/dataset/GRACEFO_L2_GFZ_MONTHLY_0063, and https://gracefo.jpl.nasa.gov, and https://cost-g.org.

**Code availability**

Python and the Climate Data Operators (CDO) scripts were used to prepare model output, climate forcing, and GRACE data. The code can be accessed at Zenodo (https://doi.org/10.5281/zenodo.15838299).



## Author contributions

F.H. performed the research and wrote the paper. All authors discussed the results and commented on the manuscript.

## Conflict of Interest Disclosure

The authors declare there are no conflicts of interest for this manuscript.