# Peer review of "Emerging global freshwater challenges unveiled through observation-constrained projections"

_EGUsphere, 2025_

## Referee Comment (RC2)

**Review for „Emerging global freshwater challenges unveiled through observation-constrained projections" by Fei Huo et al.**

*General comments:*

The authors use the EC (emergent constraint) methodology to observationally constrain TWS changes in ISIMIP3b and ISIMIP2b model output. The results show a substantial reduction in TWS when applying the constraints under different emission scenarios by the end of the century as compared to the raw model output, indicating that unconstrained model results might underestimate future water scarcity.

While the overall findings seem reasonable to me, and pointing out shortcomings of TWS model projections is a relevant topic, I have some major concerns about this study:

1. The EC method which is the backbone of the study does not get clear to me from the description in the manuscript. The text is lacking a clear introduction of the general idea behind the EC framework. Furthermore, advantages compared to other methods are not discussed.
2. The study uses ISIMIP3b data, and ISIMIP2b data as "validation" data set. However, it is not explained in which way ISIMIP2b can be used for validation. In my opinion, the EC method is applied to both data sets, and no real validation has been carried out. In combination with the unclear description of the EC approach this leaves me with doubts about the validity and robustness of the results.
3. The investigation of underlying physical processes (Section 3.3) is not convincing to me. It must be extended and discussed in more detail.
4. The authors do not discuss limitations of the EC method but see the challenges only in structural dependencies among climate models. Furthermore, uncertainties and limitations in the observational data and its implications on the results are not included in the discussion.

Overall, the manuscript would benefit from an independent validation of the method, a more elaborated explanation of the results, as well as a more critical discussion of the findings. Therefore, I recommend a major revision.

*Specific comments:*

Section 2.1: It is not only GRACE but also GRACE-FO data being used. Please add GRACE-FO in line 49. You claim mascons being more reliable than spherical harmonics in the second sentence, but in the third sentence you mention that you also incorporate SH solutions. That is a bit confusing. I would suggest to base the analysis on the usage of all (mascon and SH solutions), and not to split it into mascon and mascon+SH, i.e., replace Fig. 1 by Fig. S7. This would be easier to read and follow.

Line 70: "validated" What is the reasoning that ISIMIP2b can be used as a validation data set?

Line 73: Why regridding to 1x1° and not keep the 0.5° resolution?

Line 75 / line 110: "GRACE's baseline period" sounds as if there would be a commonly defined period to which results always refer to, also in other studies. But I think this baseline was chosen by the authors specifically for this study? Please reframe.

Line 78 – 85: The explanation of the EC approach is not clear to me (see my main concern).

Line 94 – 95: Please extend this explanation a bit. It only gets a bit clearer after reading section 3.3. However, as a purely statistical measure, the Spearman's rank correlation does not tell anything about the physical mechanisms behind two variables. Also, it is not clear in this paragraph which "variables" are meant.

Line 126 – 130: Please add a critical discussion on the "wet gets wetter" response you find here. There are also several studies that confirm the "wet gets wetter" paradigm only for a small percentage of the land area (e.g. Xiong et al. https://doi.org/10.5194/hess-26-6457-2022), and even Greve et al. (that you cite here) state that "Only 10.8% of the global land area shows a robust 'dry gets drier, wet gets wetter' pattern").

Line 135: What does "uniform" storage physics mean?

Fig 1: I do not understand how mid- and late-century TWS changes can be computed for GRACE/-FO observations (black crosses). This is probably because I did not fully understand the EC approach from the Methods section.

Line 176: The ISIMIP2b ensemble contains more and other models that the ISIMIP3b ensemble. Isn't this the main reason for the different result?

Line 190 – 193: The differences could also be due to different (number of) models used in the ISIMIP3b and ISIMIP2b ensemble, is that correct?

Line 195: Where does this distinction come from?

Figure 2 & 3: As far as I understand is Fig. 3d the difference between Fig. 2b and Fig. 3c. However, I do not see the big reduction in northern South America (the big red blob in Fig 2b) being reflected in the difference plot (Fig. 3d). Are you sure these are the correct plots? If so, please comment on this striking pattern in South America in Fig. 2b, and why it is not present in Fig 3c.

Fig. 3: "only regions with statistically significant positive EC correlations are shown" Please indicate the non-significant regions in another color (e.g. gray) for a better interpretation of the plots.

Line 230 – 232: I do not understand in which way Figure 4b and 4c, i.e., the correlation between precipitation and evapotranspiration and runoff, support the findings of the study, or help to better understand physical drivers. In my view you should either extend the analysis considerably or remove evapotranspiration and runoff.

Line 234 – 235: "More importantly, …" I do not understand how this conclusion can be derived from Figure 4. Please explain it more detailed.

Fig. 4d: I only see very few black hatches, does this mean all other areas are insignificant? The pattern seems to be quite distinct, therefore I wonder if the significance test might be too pessimistic?

Line 247 – 251: I find this analysis very interesting, but it should be extended a bit. Where do these differences come from? What is different in "wet" models compared to "dry" models, which processes might lead to this pattern of differences.

Line 260 – 261: Please explain the interconnection between lack of independent forcing and skewed distribution in Fig. 1 more detailed. For me, it is not straight forward.

Line 268: Maybe you can put 83 mm into perspective. As a pure number it does not tell a lot about the significance of the impact the constraining has on the projections.

*Technical corrections:*

Line 60: TWSAs (I think the A was never introduced as abbreviation)

Line 66 and line 194: remove "ref." and brackets

Line 100: typo, resent-day → present-day

Line 107, 115, 124: remove brackets

Fig 1: typo in x-Axis, Hitorical → Historical (and in Fig S1, S2, S7 accordingly).

Fig 3 caption: typo, Figuire → Figure

Line 259: "could also derive" is not a proper sentence

---

## Author Comment (AC1)

Revision of the manuscript " Emerging global freshwater challenges unveiled through observation-constrained projections" by Fei Huo, Yanping Li, Zhenhua Li.

General Comment.

In this study, the authors provide an analysis of future terrestrial water storage (TWS) projections using the emergent constraint (EC) methodology. The study employs multiple model ensembles (ISIMIP2b and ISIMIP3b) and GRACE observations and shows that EC calibration reduces uncertainty and corrects biases in raw model projections. The analysis of regional patterns and underlying physical mechanisms strengthens the scientific contribution. However, revisions are needed to clarify statistical methods, explicitly acknowledge uncertainties (model dependence and observational limitations). However, despite my overall positive impression, there are several areas where the paper could be improved, particularly in terms of clarity, methodological rigor, and the interpretation of results. I suggest a major revision.

Response: Thank you for your supportive comments.

My major concerns are listed below.

(1) The concept of emergent constraint (EC) methodology is addressed, but its definition and distinction from the other types of methods could be clarified earlier in the manuscript. They should provide a clear explanation of how this method differs from other methods and why it is particularly relevant in this context.

Response: Emergent constraint is a widely used method to improve climate models' accuracy of future predictions. Specifically, the EC technique aims to reduce the often uncomfortably large spread in future projections within multi-model ensembles, thereby providing more tightly constrained estimates of variables of interest (Hall, A., Cox, P., Huntingford, C. et al. Progressing emergent constraints on future climate change. Nat. Clim. Chang. 9, 269–278 (2019). https://doi.org/10.1038/s41558-019-0436-6). Such improved and physically informed constraints are crucial for climate mitigation and adaptation policymaking. We will clarify its strength and the context to apply it in the field of terrestrial water storage (TWS) in the Introduction.

(2) Refining the structure to separate background, problem, and contribution; elaborating on the novelty relative to prior EC studies; and slightly tempering assertive language would substantially improve readability and the significance of the study.

Response: We will refine the content to emphasize the novelty and contribution of our study and use a more assertive tone to better organize the whole text.

(3) Explicitly stating statistical procedures and uncertainty treatments and better articulating the rationale behind certain methodological choices (e.g., scenario selection, regression structure) would substantially strengthen the transparency and reproducibility of the study.

Response: We will add the details of the methodology to clarify the rationale behind the EC method in the Data and methods.

(4) The authors don't address the limitations or uncertainty of the study. The study used GRACE satellite data, while satellite data can have the same biases. In addition, the observed data used to constrain the projection won't be the same in the future. Therefore, you need to state the limitations of the methods.

Response: The limitations of the EC approach and GRACE will be added to the Introduction and the Data and methods.

(5) Doesn't this method depend on the model ensemble structure? Have you looked at the other models?

Response: Yes. A large ensemble size always yields more robust results. Thus, we have included all models provided by the ISIMIP3b and ISIMIP2b projects.

(6) To improve scientific rigor and interpretive balance, I recommend (1) quantifying potential impacts of model dependence, (2) expanding on remaining uncertainties, especially regarding observational and regional limitations. With these revisions, the Discussion would more convincingly convey the robustness and practical significance of the emergent constraint findings.

Response: We will improve the Discussion by expanding on the limitations of the EC method and observations/modelling datasets used in this study.

(7) Several aspects could be improved to enhance clarity, logical structure, and the articulation of the research gap in the introduction section. The introduction covers a wide range of interconnected topics (TWS processes, climate impacts, model uncertainties, EC methodology), but the transitions between them could be smoother. Consider reorganizing into:(1) importance and role of TWS; (2) current challenges and modelling uncertainties; (3) motivation for applying EC and specific research objectives. This will make the narrative flow more naturally from context → problem → solution.

Response: We will organize the Introduction to improve its clarity.

(8) The introduction cites key works Bowman et al., 2018; Brient, 2020; Hall et al., 2019; Petrova et al., 2024; Shiogama et al., 2022) but could better emphasize what has not yet been done. The statement that "potential constraints on global mean changes in terrestrial water storage have yet to be thoroughly explored" is important; consider expanding slightly on why previous EC studies focused mainly on temperature or other hydro-climatic variables, to clarify the novelty and need for this study.

Response: We will rewrite this part to highlight our contribution compared with previous research.

(9) It might also be helpful to briefly mention recent improvements or limitations of GRACE-based datasets (continuity with GRACE-FO), since these are central to the proposed constraint.

Response: The limitations of GRACE and GRACE-FO will be added to the Introduction.

(10) The Methodological Framing in the introduction needs to be clearer. When introducing the EC concept, clarify that it is a statistical relationship across models linking an observable quantity to a future response. This helps readers unfamiliar with ECs understand its basis before applying it to TWS.

Response: We will explicitly let the audience know that the EC approach is basically statistical method to improve future predictions based on connections between historical status and future response.

(11) The final sentence (lines 44-46: "By combining the proposed EC with historical observations from GRACE satellites…") appropriately sets up the study objective but could more explicitly articulate the main research question or hypothesis.

Response: Yes. We will expand on our objective and associated scientific questions in the revised manuscript.

The choice to include all three GRACE mascon solutions (JPL, CSR, GSFC) is well justified. However, please clarify how these were combined, were the datasets were averaged, used separately to estimate uncertainty, or treated as independent realizations? Linear interpolation to fill missing months in GRACE data is acceptable but may introduce temporal bias in regions

with strong seasonal variability. Consider noting why linear interpolation was deemed sufficient. It would be helpful to specify whether scaling factors were applied to GRACE data (as is often recommended to correct for signal attenuation), and if not, to justify this choice. The recommendation for Grid Scaling is described here: https://grace.jpl.nasa.gov/data/get-data/monthly-mass-grids-land.

Response: Each GRACE product is treated as an independent realization. Linear interpolation suffices because this study focuses on long-term changes and linear trends in TWS on annual timescales, rather than on resolving seasonal variability. As for the scaling factor, we didn't apply any even though grid-scale factors are recommended for observation-model comparisons, because the objective of our work is to accurately estimate the linear trend in TWS. However, "…, the gain factors tend to be dominated by the annual cycles of land water storage variations, and may thus not be suitable to quantify trends from the GRCTellus land data." (https://grace.jpl.nasa.gov/data/get-data/jpl_global_mascons/). We will improve the pertinent content in the revised manuscript.

The rationale for using both ISIMIP3b and ISIMIP2b datasets is sound, but the selection criteria for the specific LSMs and GHMs should be made explicit. Is it due to completeness, data availability, or model diversity? The decision to merge the end of the HIST run with the beginning of SSP1-2.6 to create a continuous 2004–2023 climatology should be further justified, especially given that this can introduce discontinuities in forcing conditions. Please clarify whether ensemble averaging was performed before regression (to reduce noise) or whether individual model realizations were treated independently.

Response: We included outputs from all available models with overlapping simulations across different scenarios, which is also why we chose SSP1-2.6 to cover the entire evaluation period: most LSMs and GHMs provide outputs under SSP1-2.6. Yes. Using SSP1-2.6 can introduce additional forcing signals compared with HIST, but using SSP2-4.5 would greatly reduce the number of models available for analysis. Each ensemble member is treated as an independent realization. We will revise the manuscript to clarify these points.

The description of the EC implementation is generally clear, but could benefit from additional precision: Specify how the statistical significance of the x–y relationships was assessed. Clarify whether the regression between (x) and (y) was performed globally or per grid cell, as this strongly affects the interpretation and robustness of results. Indicate the sample size (number of models) used in the regression, since ensemble size influences the confidence of emergent relationships. It would be useful to mention whether the linear assumption in the EC regression was tested. When applying the EC to calibrate projections, please specify whether bias correction

was applied before regression, and how uncertainty from both observational and model sources was propagated through the EC correction.

Response: We will add the pertinent information on statistical significance, sample size, and the linear assumption to the revised manuscript. Residual plots will be included to check linearity. The regression was conducted globally. Since the purpose of the EC method is to correct model biases using observations, no additional bias correction was applied. Both observational and model spreads influence the future EC correction, with larger spreads indicating greater uncertainty in future projections (i.e., a wider range of outcomes).

The validation step using "six driest and six wettest models" is interesting but could be more systematically described. Were these classifications based on percentiles of TWS climatology, or another quantitative criterion? Also, it might be useful to test whether "dry" and "wet" subsets produce statistically distinct EC relationships.

Response: The "dry" and "wet" models were categorized by ranking the absolute values of their TWS climatology, which is similar to a percentile-based criterion. We will use different criteria (for example, percentiles of TWS climatology) for classifications instead to test the sensitivity of the results. Furthermore, we will add the EC results obtained only using the "dry" or only the "wet" model subsets to the supplementary information, where interested readers can examine them in detail.

I have concerns regarding the statistical Strength and Presentation of the EC Relationship. The reported "Significant positive correlations ($R > 0.99$ for both ISIMIP2b and ISIMIP3b models) are found between historical and late century annual area-weighted global mean TWSAs, irrespective of the emissions scenario" appear high. Please clarify whether this reflects ensemble-mean correlations (i.e., correlation of means across models) or model-level correlations across realizations. Indicate whether statistical significance was adjusted for the number of ensemble members or grid points.

Response: Our results are consistent with previous research, which found a high correlation coefficient ($R = 0.98$) between historical and future climatology of the longest annual dry spell (LAD) based on CMIP5 and CMIP6 ensembles (Extended Data Fig.4, Petrova, I.Y., Miralles, D.G., Brient, F. et al. Observation-constrained projections reveal longer-than-expected dry spells. Nature 633, 594–600 (2024). https://doi.org/10.1038/s41586-024-07887-y). Such a high correlation in LAD, a key drought indicator, implies a global "dry-model-gets-drier" relationship in climate model simulations. This relationship can propagate into LSMs and GHMs through climate forcing. R was calculated using model-level realizations instead of ensemble means to increase the sample size. Statistical significance of correlations was adjusted for the number of ensemble members (i.e., 25 for ISIMIP3b and 31 for ISIMIP2b).

The authors note the use of both ISIMIP2b and ISIMIP3b ensembles, but it remains unclear whether the emergent relationship was derived separately for each dataset or jointly across all models. Explicitly stating this would improve transparency.

Response: The EC corrections were applied separately to the ISIMIP2b and ISIMIP3b datasets.

The authors state, "Furthermore, the EC correction constrains the discrepancies of late century TWS changes by 63% for the SSP1-2.6 scenario and 69% for the SSP3-7.0 scenario". This reduction should be quantitatively defined if it decreases in variance, range, or standard deviation. and accompanied by confidence intervals or bootstrap uncertainty ranges.

Response: We will revise the manuscript to examine TWS changes in more detail.

The Interpretation and Physical Plausibility of the physical explanation are not clear. The author states that the results are consistent with the "wet gets wetter" paradigm, but the physical explanation could be expanded to discuss exceptions, regions, or model subsets where this relationship does not hold.

Response: We will expand our analysis in the mechanism part, especially focusing on regions where this paradigm does not hold.

The statement, in lines 169-170, that EC-corrected results "produce more robust projections than conventional approaches," is reasonable but somewhat strong. The authors acknowledge in lines 172-174 that the discrepancy arises because GRACE products used for EC correction are all derived from a single source, the GRACE satellites, resulting in an underestimated uncertainty for global mean TWS. This is an important caveat and should be emphasized more explicitly as a potential limitation rather than a parenthetical remark.

Response: The limitations will be acknowledged explicitly in the revised version.

Need to describe Figure 2 clearly. Line 186, you describe Figure 2a as the TWSAs, but in line 189, you use "TWS changes". But in the figure caption, you use "ensemble averages of TWS changes". Figures 2 and 3 are central to the paper, but the text and the caption describing them could be clearer. Indicate explicitly how the EC calibration modifies the spatial distribution. The finding that only ~26% of land areas show significant EC correlations ($R > 0$, $p < 0.05$) is interesting but somewhat low; consider discussing why many regions do not exhibit a significant relationship.

Response: We will improve the clarity of the captions for Figures 2 and 3. The discussion about how the EC correction changes the spatial pattern will also be added. Although the fraction of land areas with significant EC correlations is ~26%, this value is comparable to that reported in a previous study, which used the EC calibration on the longest annual dry spell (Petrova, I.Y., Miralles, D.G., Brient, F. et al. Observation-constrained projections reveal longer-than-expected dry spells. Nature 633, 594–600 (2024). https://doi.org/10.1038/s41586-024-07887-y). This limited spatial extent likely reflects the complexity of the hydrological response to global warming. Despite the globally identified "wet gets wetter" atmospheric response in general circulation models (GCMs), the terrestrial water cycle is governed by more intricate mechanisms. In many regions, freshwater variability is strongly influenced by natural variability, including oscillations between dry and wet periods driven by El Niño, La Niña and other climate modes.

The overestimation of TWSA in northern midlatitudes and underestimation in humid regions aligns with prior findings, but it would be useful to quantify regional biases. The correlation between precipitation and TWSA changes (Fig. 4) supports the physical validity of the EC, but the analysis might benefit from partial correlation tests to control for precipitation when relating TWSA to other drivers (e.g., evapotranspiration or runoff).

Response: We will quantify regional biases and conduct partial correlation analysis to get insights to understand mechanisms of TWS changes.

The "wet vs. dry model" comparison is intriguing, but the classification criteria are somewhat arbitrary; consider clarifying or showing that results are insensitive to the chosen threshold. In discussing the "wet gets wetter" mechanism, it would be useful to acknowledge nonlinear land surface feedback (groundwater depletion, vegetation response) that may dampen or reverse this pattern regionally.

Response: We will use different criteria (for example, percentiles of TWS climatology) for classifications instead to test the sensitivity of the results. The nonlinear interactions among different land components will also be acknowledged.

Several areas require further clarification or refinement to improve balance, transparency, and interpretive rigor in the discussion section. The authors identify that the reliability of proposed ECs could be compromised due to the lack of independence among climate models (Brient, 2020; Caldwell et al., 2014). However, the discussion could be more quantitative. The statement that "diversity in global models and their climate forcings is critical" is well-taken but could be

strengthened by suggesting specific strategies, for instance, promoting structural independence in ISIMIP protocols or incorporating multi-forcing experiments to test robustness.

Response: The discussion will be refined to be more interpretive.

The reported "average TWS decrease of roughly 83 mm" is a key quantitative finding, but should be contextualized: what fraction of total terrestrial water storage does this represent? And how does this compare to previous assessments? The phrase "elevates the risk of basins being underprepared" introduces a policy implication that is not directly analyzed in the study. Consider softening this to "may imply that current water resource planning could underestimate potential shortages," unless explicit basin-level analyses were performed.

Response: We will improve this part based on the reviewer's comment.

The conclusion could be better with the limitations and broader implications. While model dependence and structural similarity are mentioned, other potential limitations deserve brief acknowledgment, such as short observational baselines, GRACE uncertainty, and non-stationary relationships under extreme forcing scenarios. The conclusion could better distinguish between confidence in the global-scale EC relationship (which appears robust) and uncertainty in regional-scale projections, where the EC significance covers only ~26% of land areas. This nuance would enhance interpretive caution. The section might benefit from a short paragraph linking the EC findings to future modelling priorities.

Response: Many thanks. We will expand the discussion of limitations from broader perspectives and analyze the uncertainty on both global and regional scales. Also, future modelling priorities will be outlined in the concluding section.

Specific comments:

The abstract mentions "low- and high-end forcing scenarios" without naming them explicitly (e.g., SSP1-2.6 and SSP3-7.0), which would aid clarity and reproducibility.

Response: The information will be added to the abstract.

Line 28-31: It is not always the case, as the warming climate can modify precipitation patterns and lead to floods in some regions. So, saying that "these changes exacerbate freshwater scarcity" is not always true for all regions.

Response: We will rephrase the text to ensure an accurate statement.

Line 38: What is the motivation for using the "emergent constraint (EC) approach"

Response: Discrepancies exist in future TWS projections, due to various factors, including uncertainties in climate forcing, the absence of key components such as surface water storage, groundwater storage, and human interventions in most land surface models (LSMs), as well as limited storage capacities within both LSMs and global hydrological models (GHMs). The EC technique, a relative novel method, is an efficient way to constrain uncertainties in future projections. Specifically, the EC technique aims to reduce the often uncomfortably large spread in future projections within multi-model ensembles, thereby providing more tightly constrained estimates of variables of interest (Hall, A., Cox, P., Huntingford, C. et al. Progressing emergent constraints on future climate change. Nat. Clim. Chang. 9, 269–278 (2019). https://doi.org/10.1038/s41558-019-0436-6). Such improved and physically informed constraints are crucial for climate mitigation and adaptation policymaking. We will modify the introduction based on your comments.

Line 45: The phrase "successfully constrain future TWS changes could be rephrased to avoid implying confirmed success, e.g., "apply the EC framework to constrain projections of future TWS changes."

Response: Modified as suggested.

Check that reference years (e.g., GRACE citations) correspond to the latest data versions.

Response: Modified as suggested.

Line 49: Why did you choose the period 2004-2023? Why not consider a long period from [Humphrey, V., & Gudmundsson, L. (2019). GRACE-REC: a reconstruction of climate-driven water storage changes over the last century. Earth System Science Data, 11(3), 1153-1170.] https://figshare.com/articles/dataset/GRACE-REC_A_reconstruction_of_climate-driven_water_storage_changes_over_the_last_century/7670849

Response: GRACE-REC is a widely used reconstructed dataset, but as recommended by the authors of Humphrey, V., & Gudmundsson, L. (2019), "the reconstructed TWS trends mainly depend on the trends initially present in the driving precipitation data", and "it should be clear that there will be differences between the trends found in GRACE and the trends found in the reconstruction. Such discrepancies are expected because the reconstruction does not represent

several sources of long-term changes in TWS…". Therefore, we didn't use GRACE-REC in our study because capturing linear trends are crucial to our analysis.

Line 49-57: The authors don't state the name of the variable used in the study in all paragraphs of the Observation section, which is important for the readers.

Response: Modified as suggested.

Line 56-57: Give more explanation regarding the linear interpolation model. The authors need to convince me that this approach is acceptable. Why not consider the Humphrey, V., & Gudmundsson, L. (2019), which is a reconstruction of data.

Response: Our study focuses on long-term changes and linear trends in TWS on annual timescales, rather than on resolving seasonal variability. As we mentioned above, GRACE-REC is more suitable for analysis of interannual variability. The author of Humphrey, V., & Gudmundsson, L. (2019) stated, and I quote: "the trends in GRACE-REC cannot be directly evaluated against the trends from GRACE itself."

Line 62: What are those five general circulation models (GCMs)?

Response: We will improve clarity in the revised manuscript.

Line 63: Ensure consistent notation: SSP should be defined at first use in the main text.

Response: Modified as suggested.

Line 73: Which method have you used to regrid the data?

Response: Details added.

Line 74: Why was the comparison only made relative to the period 2004-2009? Is it no too short?

Response: The 2004-2009 baseline time was used solely to calculate monthly climatologies and corresponding anomalies. As long as this baseline (climatology) is fixed (i.e., constant at a given grid point), its short duration doesn't affect the results because our focus is on long-term changes and linear trends of these anomalies. We will clarify this in the revised version.

Line 100: What does "resent-day global mean TWSAs"? Is it a typo: "resent-day" → "present-day."

Response: Modified as suggested.

Lines 106-107: Ensure consistent reference formatting and in-text citation style (e.g., "(for example, (Cai et al., 2025; …)" should remove nested parentheses).

Response: Modified as suggested.

Line 168-170: Phrases such as "produce more robust projections" and "further exacerbating existing water stress worldwide" are somewhat strong. Consider tempering the language to reflect the uncertainty and limitations of observational constraints.

Response: Modified as suggested.

-Fig S1 caption: In the legend of Fig S1a, you describe the crosses for observation, but you don't state it in the Fig S1 caption, "Dots and crosses represent global averages of TWSAs from ensemble members." As same as for FigS2.

Response: Modified as suggested.

It will be convenient to add the observation in Fig. S1.b to make it for comparison. -It will be better to have a unique color bar for the figS4 to represent the difference. For (a) and (b), you used red to represent the decrease in TWS and precipitation, while it is blue for the Evapotranspiration and Total runoff. The same for FigS5, FigS6.

Response: Modified as suggested.

-There is only 5 crosses from observation in the Fig. S7, while you declare that you replaced the mascon solutions with 7 Grace-derived TWSA.

Response: Some datasets show nearly identical values (e.g., -13.10 for JPLM and -13.72 for GSFC; -9.61 for JPL and -9.47 for GFZ), causing their crosses to overlap in the figure.

- Be consistent with the format of supplementary figure citation in the main manuscript, for example, in line 179, you use "Supplementary Fig. 7" but in line 180 it is "Fig. S2". And make Sure that all supplementary figures are addressed in the main manuscript.

Response: Modified as suggested.

---

## Author Comment (AC2)

Review for „Emerging global freshwater challenges unveiled through observation-constrained projections" by Fei Huo et al.

General comments:

The authors use the EC (emergent constraint) methodology to observationally constrain TWS changes in ISIMIP3b and ISIMIP2b model output. The results show a substantial reduction in TWS when applying the constraints under different emission scenarios by the end of the century as compared to the raw model output, indicating that unconstrained model results might underestimate future water scarcity. While the overall findings seem reasonable to me, and pointing out shortcomings of TWS model projections is a relevant topic, I have some major concerns about this study:

Response: We appreciate your supportive assessment of our paper.

1. The EC method which is the backbone of the study does not get clear to me from the description in the manuscript. The text is lacking a clear introduction of the general idea behind the EC framework. Furthermore, advantages compared to other methods are not discussed.

Response: Discrepancies exist in future TWS projections, due to various factors, including uncertainties in climate forcing, the absence of key components such as surface water storage, groundwater storage, and human interventions in most land surface models (LSMs), as well as limited storage capacities within both LSMs and global hydrological models (GHMs). The EC technique, a relative novel method, is an efficient way to constrain uncertainties in future projections. Specifically, the EC technique aims to reduce the often uncomfortably large spread in future projections within multi-model ensembles, thereby providing more tightly constrained estimates of variables of interest (Hall, A., Cox, P., Huntingford, C. et al. Progressing emergent constraints on future climate change. Nat. Clim. Chang. 9, 269–278 (2019). https://doi.org/10.1038/s41558-019-0436-6). Such improved and physically informed constraints are crucial for climate mitigation and adaptation policymaking. We will modify the introduction based on your comments.

2. The study uses ISIMIP3b data, and ISIMIP2b data as "validation" data set. However, it is not explained in which way ISIMIP2b can be used for validation. In my opinion, the EC method is applied to both data sets, and no real validation has been carried out. In combination with the unclear description of the EC approach this leaves me with doubts about the validity and robustness of the results.

Response: ISIMIP3b and ISIMIP2b data were no validation datasets but necessary parts for the EC method, used to identify statistically significant relationships between annual global mean

changes in TWS (y) and historical annual global mean TWSA climatologies (x), across a variety of global models. Such relationship (regression between y and x) along with the GRACE observations were then used to calibrate future TWS changes to reduce the spread in future projections within ISIMIP3b and ISIMIP2b models. Specifically, by replacing x with the GRACE observations in the regression, we obtained the calibrated future changes in TWS.

3. The investigation of underlying physical processes (Section 3.3) is not convincing to me. It must be extended and discussed in more detail.

Response: We will quantify regional biases and apply partial correlation analysis among different variables to get insights to understand mechanisms of TWS changes.

4. The authors do not discuss limitations of the EC method but see the challenges only in structural dependencies among climate models. Furthermore, uncertainties and limitations in the observational data and its implications on the results are not included in the discussion.

Response: The applicability of the EC technique is inherently limited by the knowledge space represented by the ensemble of climate models. If key physical processes are oversimplified or absent in the models, the EC method cannot identify or constrain those processes. Thus, the inter-model spread captured by the EC relationship may be unrealistic or unjustified due to the lack of a sound physical basis. We will improve the conclusion so that it could be better with the limitations and broader implications, including brief acknowledgment of potential limitations, such as short observational baselines, GRACE uncertainty, and non-stationary relationships under extreme forcing scenarios.

Overall, the manuscript would benefit from an independent validation of the method, a more elaborated explanation of the results, as well as a more critical discussion of the findings. Therefore, I recommend a major revision.

Response: We will enhance the manuscript by improving the methodology, the presentation of results, and the discussion.

Specific comments:

Section 2.1: It is not only GRACE but also GRACE-FO data being used. Please add GRACE-FO in line 49. You claim mascons being more reliable than spherical harmonics in the second sentence, but in the third sentence you mention that you also incorporate SH solutions. That is a bit confusing. I would suggest to base the analysis on the usage of all (mascon and SH solutions),

and not to split it into mascon and mascon+SH, i.e., replace Fig. 1 by Fig. S7. This would be easier to read and follow.

Response: We will modify the text to improve its clarity. Since the inclusion of SH solutions has little impact on the outcomes, we have decided to remove the related discussion from the main text while keeping Fig. S7 for readers interested in the results that include SH solutions.

Line 70: "validated" What is the reasoning that ISIMIP2b can be used as a validation data set?

Response: The reviewer may have been misled by the use of this term "validated". As we mentioned in the response to general comment #2, ISIMIP3b and ISIMIP2b data were no validation datasets but necessary parts for the EC method, used to identify statistically significant relationships between annual global mean changes in TWS (y) and historical annual global mean TWSA climatologies (x), across a variety of global models. Therefore, we will improve the clarity in the revised version.

Line 73: Why regridding to 1x1° and not keep the 0.5° resolution?

Response: Some SH solutions such as GFZ RL06 and COST-G RL01 provide data only at 1-degree resolution. Thus, we regridded all datasets onto a common grid.

Line 75 / line 110: "GRACE's baseline period" sounds as if there would be a commonly defined period to which results always refer to, also in other studies. But I think this baseline was chosen by the authors specifically for this study? Please reframe.

Response: We will update the relevant content.

Line 78 – 85: The explanation of the EC approach is not clear to me (see my main concern).

Response: We will improve its clarity. Please refer to our response to general comment #2.

Line 94 – 95: Please extend this explanation a bit. It only gets a bit clearer after reading section 3.3. However, as a purely statistical measure, the Spearman's rank correlation does not tell anything about the physical mechanisms behind two variables. Also, it is not clear in this paragraph which "variables" are meant.

Response: We will clarify this and dig deeper about the physical mechanism using other statistical methods such as partial correlations in the revised manuscript.

Line 126 – 130: Please add a critical discussion on the "wet gets wetter" response you find here. There are also several studies that confirm the "wet gets wetter" paradigm only for a small percentage of the land area (e.g. Xiong et al. https://doi.org/10.5194/hess-26-6457-2022), and even Greve et al. (that you cite here) state that "Only 10.8% of the global land area shows a robust 'dry gets drier, wet gets wetter' pattern").

Response: We agree that this "wet gets wetter" signal, primarily identified from global climate models may be obscured when multiple climate drivers and natural variability are considered. Hence, our original conclusions were limited to the agreement of "this 'wet gets wetter' signal over water-sufficient lands". However, evidence from Greve et al. and Xiong et al. shows that differences in study periods, datasets, and the variables used to measure hydrological conditions can substantially influence the inferred responses. Therefore, we will expand the discussion to include arguments both supporting and questioning the "wet gets wetter" response.

Line 135: What does "uniform" storage physics mean?

Response: We don't assume identical storage physics across different LSMs/GHMs. We will revise this part to clarify this.

Fig 1: I do not understand how mid- and late-century TWS changes can be computed for GRACE/-FO observations (black crosses). This is probably because I did not fully understand the EC approach from the Methods section.

Response: This scatterplot is a conventional way to illustrate the EC relationship. When observations are considered, the focus is typically on the information shown on the x-axis, which is why the mean of observations is indicated by a black vertical line (as it has no corresponding y-axis values). Similarly, the box plots show no information along the x-axis.

Line 176: The ISIMIP2b ensemble contains more and other models that the ISIMIP3b ensemble. Isn't this the main reason for the different result?

Response: Different suites of models and ensemble members can influence the EC-corrected results, which is why we included as many models as possible. We will acknowledge uncertainties in our results that may arise from differences in the suite of models and ensemble members.

Line 190 – 193: The differences could also be due to different (number of) models used in the ISIMIP3b and ISIMIP2b ensemble, is that correct?

Response: Yes. We will acknowledge uncertainties in our results that may be introduced by the differences in the suite of models and ensemble members at the end of section 3.1.

Line 195: Where does this distinction come from?

Response: We will modify the sentence to improve clarity.

Figure 2 & 3: As far as I understand is Fig. 3d the difference between Fig. 2b and Fig. 3c. However, I do not see the big reduction in northern South America (the big red blob in Fig 2b) being reflected in the difference plot (Fig. 3d). Are you sure these are the correct plots? If so, please comment on this striking pattern in South America in Fig. 2b, and why it is not present in Fig 3c.

Response: Yes, Fig. 3d represents the difference between the two panels (Fig. 2b minus Fig. 3c). We have double-checked the figure, and it is correct. We will also add another figure to the supplementary information. This new figure is nearly identical to Fig. 3, but the EC correction is applied to all grid cells. It is obvious in Fig. S8c that the aforementioned big red blob remains.

[Figure]

Fig. S8 Same as Figure 3 but the EC correction is applied to all grid cells.

Fig. 3: "only regions with statistically significant positive EC correlations are shown" Please indicate the non-significant regions in another color (e.g. gray) for a better interpretation of the plots.

Response: The colormap used in Fig. 3 is "'RdBu'" (without a white midpoint). Accordingly, all regions shown in white represent grid cells where the results are not statistically significant. We will revise the figure caption to clarify this.

Line 230 – 232: I do not understand in which way Figure 4b and 4c, i.e., the correlation between precipitation and evapotranspiration and runoff, support the findings of the study, or help to better understand physical drivers. In my view you should either extend the analysis considerably or remove evapotranspiration and runoff.

Response: We will expand our analysis in the revised manuscript.

Line 234 – 235: "More importantly, …" I do not understand how this conclusion can be

derived from Figure 4. Please explain it more detailed.

Response: We will conduct partial correlation analysis among different variables to identify true associations between TWS and other variables such as precipitation, evapotranspiration, and runoff.

Fig. 4d: I only see very few black hatches, does this mean all other areas are insignificant? The pattern seems to be quite distinct, therefore I wonder if the significance test might be too pessimistic?

Response: Black hatches indicate statistically significant differences at the 5% level, as determined by Welch's t-test. A permutation test with 100 random permutations was conducted to estimate the p-values. We also tested a more relaxed threshold ($p < 0.1$), but the significance patterns were identical. We infer that large inter-model variability in TWS among "wet" and "dry" models may lead to statistical non-significance in some regions, even where the differences (wet minus dry models) are substantial.

Line 247 – 251: I find this analysis very interesting, but it should be extended a bit. Where do these differences come from? What is different in "wet" models compared to "dry" models, which processes might lead to this pattern of differences.

Response: We will compare regarding the physical processes and model components of the "wet" with "dry" model groups to provide more insight.

Line 260 – 261: Please explain the interconnection between lack of independent forcing and skewed distribution in Fig. 1 more detailed. For me, it is not straight forward.

Response: Previous studies have applied the EC approach to constrain future projections of the longest annual dry spell (LAD) climatology using CMIP5 and CMIP6 ensembles (Petrova, I.Y., Miralles, D.G., Brient, F. et al. Observation-constrained projections reveal longer-than-expected dry spells. Nature 633, 594–600 (2024). https://doi.org/10.1038/s41586-024-07887-y). That study showed approximately Gaussian distributions in the raw (pre-EC) projections of CMIP5 and CMIP6 ensembles (Fig. 2). In contrast, the skewness observed in the raw projections in our study may stem from the lack of independent forcing across LSMs/GHMs in the ISIMIP framework. However, this interpretation is speculative and requires further supporting evidence. We will therefore revise this section to rely only on more robust results from our analysis.

Line 268: Maybe you can put 83 mm into perspective. As a pure number it does not tell a lot about the significance of the impact the constraining has on the projections.

Response: We will rewrite this section based on the reviewer's comments.

Technical corrections:

Line 60: TWSAs (I think the A was never introduced as abbreviation)

Line 66 and line 194: remove "ref." and brackets

Line 100: typo, resent-day → present-day

Line 107, 115, 124: remove brackets

Fig 1: typo in x-Axis, Hitorical → Historical (and in Fig S1, S2, S7 accordingly).

Fig 3 caption: typo, Figuire → Figure

Line 259: "could also derive" is not a proper sentence

Response: All typos will be corrected in the revised manuscript.